# Directly Aligning the Full Diffusion Trajectory with Semantic Relative Preference Optimization

## Abstract

Recent studies have demonstrated the effectiveness of aligning diffusion models with human preferences using differentiable reward. However, they exhibit two primary challenges: (1) they rely on multistep denoising with gradient computation for reward scoring, which is computationally expensive, thus restricting optimization to only a few diffusion steps; (2) they often need continuous offline adaptation of reward models in order to achieve desired aesthetic quality, such as photorealism or precise lighting effects. To address the limitation of multistep denoising, we propose Direct-Align, a method that predefines a noise prior to effectively recover original images from any time steps via interpolation, leveraging the equation that diffusion states are interpolations between noise and target images, which effectively avoids over-optimization in late timesteps. Furthermore, we introduce Semantic Relative Preference Optimization (SRPO), in which rewards are formulated as text-conditioned signals. This approach enables online adjustment of rewards in response to positive and negative prompt augmentation, thereby reducing the reliance on offline reward fine-tuning. By fine-tuning the FLUX.1.dev model with optimized denoising and online reward adjustment, we improve its human-evaluated realism and aesthetic quality by over 3x.

## 1 Introduction

Online reinforcement learning (Online-RL) (Xu et al., 2023; Clark et al., 2023; Prabhudesai et al., 2024) methods that perform a direct gradient update through differentiable rewards have demonstrated substantial potential to align diffusion models with human preferences. Compared to policy-based approaches (Fan et al., 2023; Fan & Lee, 2023; Black et al., 2023; Xue et al., 2025; Liu et al., 2025; Wang et al., 2025), these methods use analytical gradients rather than policy gradients, allowing more efficient training. Despite their promising performance, these methods are frequently observed to introduce artifacts after training, such as oversaturation or unrealistic textures. These issues are mainly attributable to two primary limitations: First, they restrict optimization to only a few diffusion steps, making them more susceptible to reward hacking, a phenomenon where models achieve high reward scores for low-quality images (Liu et al., 2025; Clark et al., 2023; Domingo-Enrich et al., 2024; Xue et al., 2025; Lee et al., 2023; Pan et al., 2022). Second, they lack an mechanism to adjust rewards and require costly offline preparations to tune for desired aesthetic such as photo-realism or precise lighting.

The first limitation stems from the conventional process of aligning the generation progress with a reward model. Existing methods typically backpropagate gradients through a standard multistep sampler (Ho et al., 2020; Song et al., 2020a), such as DDIM. However, these frameworks are not only computationally expensive, but also prone to severe optimization instability, such as gradient explosion. This issue becomes particularly acute when backpropagating gradients through the long computational graphs of early diffusion timesteps, forcing these methods to restrict optimization to the later stages of the trajectory. However, this narrow focus on late-stage timesteps makes the model prone to overfitting the reward, as demonstrated in our experiment (see fig. 5). This overfitting manifests as reward hacking, leading models to exploit known biases in popular reward models. For instance, HPSv2 (Wu et al., 2023) develops a preference for reddish tones, PickScore (Kirstain et al., 2023) for purple images and ImageReward (Xu et al., 2023) for overexposed regions. Previous

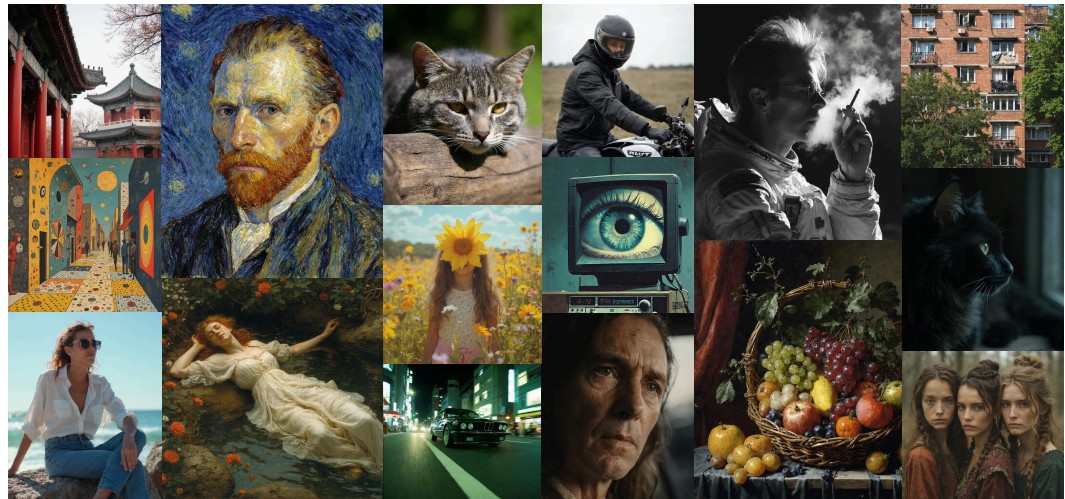

Figure 1: **Images generated by FLUX.1-dev finetuned our Semantic Relative Preference Optimization (SRPO)** Our method offers both impressive realism and aesthetic quality, and can be trained in only 10 minutes with 32 NVIDIA H20 GPUs, demonstrating exceptional efficiency.

work (Ba et al., 2025; Lee et al., 2025) has also found that these models tend to prefer smoothed images with low-detail. To address this limitation, our method first injects predefined noise into the clean image, enabling the model to directly interpolate back to the original from any given timestep.

The second challenge is the absence of mechanisms for online reward adjustment to accommodate the evolving needs of real-world scenarios. Both the research community and industry often make offline adjustments before RL. For example, contemporaneous works such as ICTHP (Ba et al., 2025) and Flux.1 Krea (Lee et al., 2025) have shown that existing reward models tend to favor images with low aesthetic complexity. ICTHP addresses this issue by collecting a large, high-quality dataset to fine-tune the reward model, while other works such as DRaFT (Clark et al., 2023) and DanceGRPO (Xue et al., 2025) search for suitable reward systems to modulate image attributes such as brightness and saturation. In contrast, we propose treating rewards as text-conditional signals, enabling online adjustment through prompt augmentation. To further mitigate reward hacking, we regularize the reward signal by using the relative difference between conditional reward pairs, defined by predefined positive and negative keywords applied to the same sample, as the objective function. This approach effectively filters out information irrelevant to semantic guidance. Consequently, we introduce Semantic Relative Preference Optimization (SRPO), built upon Direct-Align.

In our experiments, we first leverage SRPO to adjust standard reward models to two critical but often overlooked aspects: image realism and texture detail. Next, we rigorously compare SRPO with several state-of-the-art Online RL-based methods on FLUX.1.dev, including ReFL (Xu et al., 2023), DRaFT (Clark et al., 2023), DanceGRPO (Xue et al., 2025), across a diverse set of evaluation metrics such as Aesthetic predictor 2.5 (Unknown, 2025), Pickscore (Kirstain et al., 2023), ImageReward (Xu et al., 2023), GenEval (Ghosh et al., 2023), and human assessments. Remarkably, our approach demonstrates a substantial improvement in human evaluation metrics. Specifically, compared to the baseline FLUX.1.dev Labs (2024) model, our method achieves an approximate 3.7-fold increase in perceived realism and a 3.1-fold improvement in aesthetic quality. Finally, we emphasize the efficiency of our approach. By applying SRPO to the FLUX.1.dev and training for only 10 minutes on HPDv2 dataset (Wu et al., 2023), our method enables the model to surpass the performance of the latest version of FLUX.1.Krea (Lee et al., 2025) on the HPDv2 benchmark.

In summary, the key contributions are as follows:

(1) **Mitigating Reward Hacking:** The proposed framework effectively mitigates reward hacking. Specifically, it removes the limitation of previous methods that could only train on the later diffusion process. Furthermore, we introduce a Semantic Relative Preference mechanism, which regularizes the reward signal by evaluating each sample with both positive and negative preference.

(2) **Online Reward Adjustment:** We reformulate reward signals as text-conditioned preferences, which enables dynamic control of the reward model via prompt augmentation. This approach reduces the reliance on reward-system or reward-model fine-tuning.

(3) **State-of-the-Art Performance:** Extensive evaluations demonstrate that our approach achieves state-of-the-art results. Our method significantly enhances the realism of large-scale flow matching models without requiring additional data, achieving convergence within just 10 minutes of training.

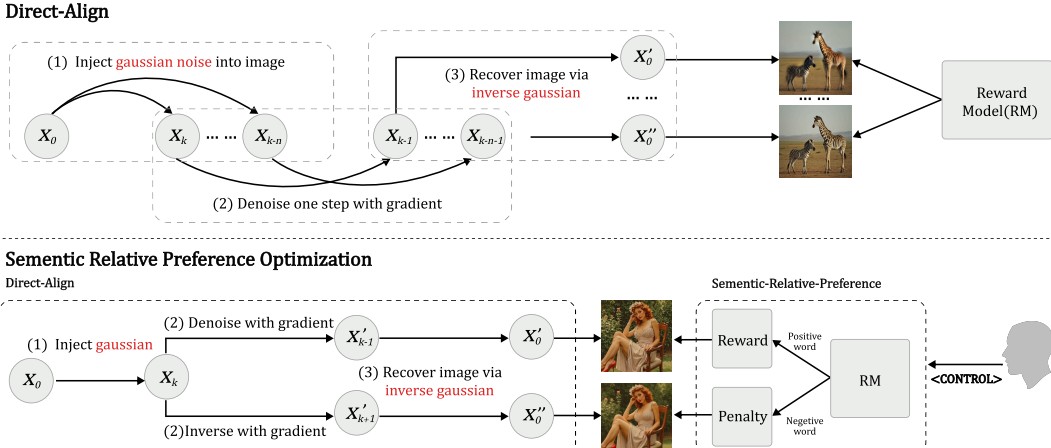

Figure 2: **Method Overview.** The SRPO contains two key elements: Direct-Align, and a single reward model that derives both rewards and penalties from positive and negative prompts. The pipeline of Direct-Align consists of four stages: (0) generate an image for training; (1) inject noise into image; (2) perform one-step denoise/inversion; (3) recover image.

## 2 RELATED WORK

**Optimization on Diffusion Timesteps.** Recent advances (Domingo-Enrich et al., 2024; Albergo et al., 2023; Ma et al., 2024; Li et al., 2024) have demonstrated that diffusion models (Song et al., 2020b;a; Ho et al., 2020) and flow matching methods (Liu et al., 2022; Lipman et al., 2022) can be unified under a continuous-time SDE/ODE framework, where images are generated through a progressive trajectory, with the early stages modeling the low-frequency structure and later steps refining high-frequency details. Recent studies (Zhang et al., 2025; Liang et al., 2025) suggest that optimizing early timesteps improves training efficiency and generation quality. However, standard direct backpropagation with reward approaches (Xu et al., 2023; Clark et al., 2023; Prabhudesai et al., 2024) struggle with early stage optimization due to excessive noise that corrupts reward gradients. To address this, we propose a novel sampling strategy that recovers clean images from highly noisy inputs in a single step, enabling effective optimization at early diffusion stages.

**Refining Reward Models for Human Preferences.** A central challenge in aligning diffusion models with human preferences is reward hacking, which often arises from a mismatch between existing reward models and genuine human preferences. This discrepancy can be attributed to two primary factors. First, modeling inherently subjective human aesthetics is a significant challenge, as illustrated by the low inter annotator agreement in previous reports (Xu et al., 2023; Ma et al., 2025): 65.7% for the ImageReward test set and 59.7% for HPDv2. Second, current reward models are typically trained on limited criteria and outdated model generations, capturing preferences only at a coarse granularity learned from their training data like *Fielidy and Text-to-image alignment* in ImageReward, and often require offline adjustment before RL to align with higher aesthetic demands. For example, ICTHP (Ba et al., 2025) highlights the bias of the reward models toward low detail and low aesthetic images, while HPSv3 (Ma et al., 2025) addresses this by training the rewards with advanced models and real images, and MPS (Zhang et al., 2024) introduces more fine-grained criteria for training. In contrast, our work focuses on how the reward signal is utilized within the RL process, employing text-conditional preference to align reward attribution with targeted attributes and filter out non-essential biases. This endows our method with robust generalization and provides

different rewards, significantly improving the visual quality of the latest FLUX.1.dev model using standard rewards like HPSv2 without requiring advanced or specifically fine-tuned alternatives.

## 3 METHOD

### 3.1 DIRECT-ALIGN

**Limitations of Existing Approaches.** Existing direct backpropagation algorithms optimize diffusion models by maximizing reward functions evaluated on generated samples. Current approaches (Xu et al., 2023; Clark et al., 2023; Prabhudesai et al., 2024) typically employ a two-stage process: (1) sampling noise without gradients to obtain an intermediate state $x_k$, followed by (2) a differentiable prediction is conducted to produce an image. This enables gradients from the reward signal to be backpropagated through the image generation process. The final objectives of these methods can be categorized into two types.

$$\text{Draft-like:} \quad r = R(\text{sample}(\mathbf{x_t}, \mathbf{c})), \tag{1}$$

$$\text{ReFL-like:} \quad r = R(\frac{\mathbf{x_t} - \sigma_t \epsilon_\theta(\mathbf{x_t}, t, \mathbf{c})}{\alpha_t}), \tag{2}$$

DRaFT (Clark et al., 2023) performs regular noise sampling throughout the process, including the final few steps and even the last step, as multistep sampling leads to significant computational cost and unstable training when the number of steps exceeds five, as reported in the original work. Similarly, ReFL (Xu et al., 2023) also opts for a later value of $k$ before performing a one-step prediction to obtain $x_0$, as the one-step prediction tends to lose accuracy at early timestep. Both methods restrict the reinforcement learning process to the later stages of sampling.

**Single-Step Image Recovery.** To address the limitation mentioned above, an accurate single-step prediction is essential. Our key insight is inspired by the forward formula in diffusion models, which suggests that a clean image can be reconstructed directly from an intermediate noisy image and Gaussian noise as shown in eq. (4). Building on this insight, we propose a method that begins by injecting ground-truth Gaussian noise prior into an image, placing it at a specific timestep $t$ to initiate optimization. A key advantage of this approach is the existence of a closed-form solution, derived from Eq. 4, which can directly recover the clean image from this noisy state. This analytical solution obviates the need for iterative sampling, thus avoiding its common pitfalls, such as gradient explosion, while preserving high accuracy even at early high-noise timesteps (see fig. 5).

$$\mathbf{x_t} = \alpha_t \mathbf{x_0} + \sigma_t \epsilon_{gt}, \tag{3}$$

$$\mathbf{x_0} = \frac{\mathbf{x_t} - \sigma_t \epsilon_{gt}}{\alpha_t}, \tag{4}$$

As shown in Eqs. 2-5, our method combines ground-truth vectors and model predictions to denoise the noisy image. The contributions of the ground-truth and predicted components are weighted by $\Delta\sigma_t$ and $\sigma_t - \Delta\sigma$, respectively.

$$r = r(\frac{\mathbf{x_t} - \Delta\sigma_t \epsilon_\theta(\mathbf{x_t}, t, \mathbf{c}) - (\sigma_t - \Delta\sigma)\epsilon}{\alpha_t}), \tag{5}$$

**Reward Aggregation Framework.** Our framework (fig. 2) generates clean images $x_0$ and injects noise in a single step. For enhanced stability, we perform multiple noise injections to produce a sequence of images $\{x_k, \ldots, x_{k-n}\}$ from the same $x_0$. Subsequently, we apply denoising and recovery processes to each image in the sequence, allowing for the computation of intermediate rewards. These rewards are then aggregated using a decaying discount factor through gradient accumulation, which helps mitigate reward hacking at later timesteps.

$$r(\mathbf{x_t}) = \lambda(t) \cdot \sum_k^{k-n} r(x_i - \epsilon_\theta(\mathbf{x_i}, i, \mathbf{c}), \mathbf{c}), \tag{6}$$

## 3.2 SEMANTIC-RELATIVE PREFERENCE OPTIMIZATION

**Semantic Guided Preference.** Modern Online-RL for text-to-image generation employs reward models to evaluate output quality and guide optimization. These models typically combine an image encoder $f_{img}$ and text encoder $f_{txt}$ to compute similarity, following the CLIP architecture (Radford et al., 2021). In our experiments, we observe that the reward can be interpreted as an image-dependent function parameterized by a text embedding denoted as $C$. Crucially, we find that strategically augmenting the prompts $p$ with magic control words denoted as $p_c$ can steer the reward by modifying the semantic embedding, therefore we propose the Semantic Guided Preference (SGP) that shifts reward preference by text condition.

$$r_{SGP}(\mathbf{x}) = RM(\mathbf{x}, (\mathbf{p_c}, \mathbf{p})) \propto f_{img}(\mathbf{x})^T \cdot \mathbf{C}_{(\mathbf{p_c}, \mathbf{p})}, \tag{7}$$

Although this approach enables controlled preference, it still inherits the original reward model's biases. To address this limitation, we further propose the Semantic-Relative Preference mechanism.

**Semantic-Relative Preference.** Existing approaches often combine multiple reward models to prevent overfitting to any single preference signal. Although this can balance opposing biases (e.g., using CLIPScore's underexposure to offset HPSv2.1's oversaturation tendencies (Xue et al., 2025)). As shown in fig. 5, it merely adjusts reward magnitudes rather than aligning optimization directions, resulting in compromised trade-offs rather than true bias mitigation. Based on the insight that reward bias primary originates from the image branch (as the text branch does not backpropagation gradient), we introduce a technique to generate a pair of opposing reward signals from a same image through prompt augmentation, which facilitates the propagation of negative gradients for regularization. This approach effectively neutralizes general biases via negative gradients while preserving specific preferences in semantic difference.

$$r_{SRP}(\mathbf{x}) = f_{img}(\mathbf{x})^T \cdot (\mathbf{C}_1 - \mathbf{C}_2), \tag{8}$$

where $C_1$ represents desired attributes (e.g., *realistic*) and $C_2$ encodes unwanted features. This formulation explicitly optimizes for target characteristics while penalizing undesirable ones. For implementation, we simply add control phrases to prompts (e.g., `<control>. <prompt>`), maintaining the syntactic structure for scoring.

**Inversion-Based Regularization.** Compared to previous methods that rely on model-based image reconstruction and can only optimize along the denoising chain, our Direct-Align approach decouples reconstruction from the computational graph by using a fixed prior, enabling more flexible optimization. As a result, our method supports optimization in the inversion direction. We simplify the reward formulations for both directions by representing them in terms of a constant $\mathbf{K}$ and model prediction, as shown in eq. (9). Consequently, the denoising process performs gradient ascent, fitting the reward, whereas the inversion process has the opposite effect.

$$r_1 = r_1 \left( \frac{\mathbf{K} \pm \Delta \sigma_t \epsilon_\theta(\mathbf{x_t}, t, \mathbf{c})}{\alpha_t} \right), \tag{9}$$

Empirical analysis indicates that reward hacking predominantly occurs at high-frequency timesteps. By employing the inversion mechanism, we decouple the penalization term and the reward term from SRP at different timesteps, thereby enhancing the robustness of the optimization process.

## 4 EXPERIMENTS

We evaluate Online-RL algorithms using FLUX.1.dev (Labs, 2024) as our base model, a state-of-the-art open-source model, hereafter referred to as FLUX. All methods use HPS (Wu et al., 2023)(short for HPSv2.1) as the reward model and train on the Human Preference Dataset v2 (Wu et al., 2023), which contains four visual concepts from DiffusionDB (Wang et al., 2022). Direct propagation methods are run on 32 NVIDIA H20 GPUs. For DanceGRPO (Xue et al., 2025), we follow the official FLUX configurations on 16 NVIDIA H20 GPUs.

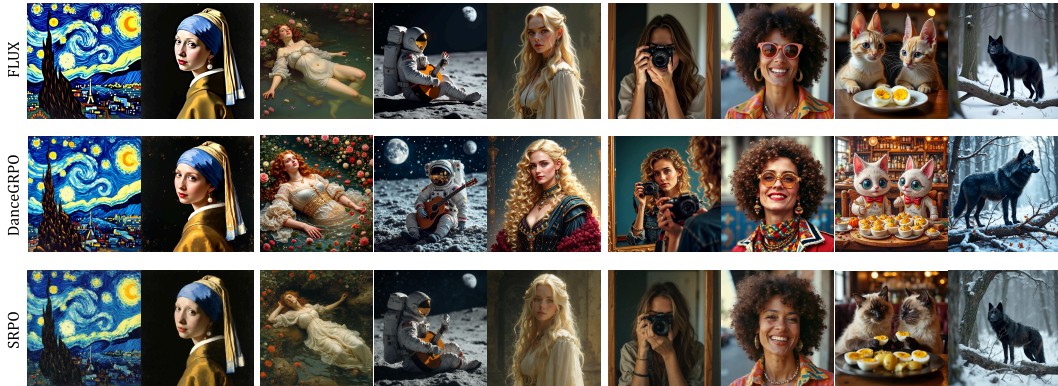

Figure 3: **Qualitative Comparison on FLUX, DanceGRPO and SRPO with same seed.** Our approach demonstrates superior performance in realism and detail complexity.

For direct propagation methods, we use 25 sampling steps to maintain gradient accuracy and 50 sampling steps during inference to ensure a fair comparison with the original FLUX.1.dev. We also compare the latest opensource FLUX.1 release from Krea (Lee et al., 2025) with our own fine-tuned FLUX.1.dev model. For Krea, we used its default configuration (28 sampling steps, Guidance 4.5).

## 4.1 EVALUATION PROTOCOL.

**Automatic metrics.** We assess image quality using established metrics on the HPDv2 benchmark (3,200 prompts). Our evaluation combines four standard measures: Aesthetic Score v2.5 (Unknown, 2025), PickScore (Kirstain et al., 2023), ImageReward (Xu et al., 2023), and HPS (Wu et al., 2023), which collectively evaluate aesthetic quality and semantic alignment. Furthermore, we introduce our SRP reward, which quantifies the difference between the score extracted by HPS from the prompts prefixed with "*Realistic photo*" ($C_1$) and "*CG Render*" ($C_2$) using HPS. For comprehensive evaluation, we employ GenEval (Ghosh et al., 2023) and DeQA (You et al., 2025).

**Human Evaluation.** We conduct a comprehensive human evaluation study comparing generative models using a rigorously designed assessment framework. The evaluation involves 10 trained annotators and 3 domain experts to ensure statistical significance and professional validation. Our data set comprises 500 prompts (first 125 prompts from each of the four subcategories in the HPD benchmark). Each prompt was evaluated by five distinct annotators in a fully crossed experimental design. The assessment focuses on four critical dimensions of image quality: (1) Text-image alignment (semantic consistency), (2) Realism and artifact presence, (3) Detail complexity and richness, and (4) Aesthetic composition and appeal. Each dimension is rated using a four-level ordinal scale: Excellent (fully meets criteria), Good (minor deviations), Pass (moderate issues), and Fail (significant deficiencies). To maintain evaluation reliability, we implement a multi-stage quality control process: (1) Experts train and calibrate annotators, (2) Systematic resolution of scoring discrepancies, and (3) Continuous validation of assessment criteria. The detail criteria in the table 2.

| Method | \multicolumn{5}{c}{Reward} | | | | | \multicolumn{2}{c}{Other Metrics} | | GPU hours(H20) |
| | Aes | Pick | ImageReward | HPS | SRP | GenEval | DeQA | |
| --- | --- | --- | --- | --- | --- | --- | --- | --- |
| FLUX | 5.867 | 22.671 | 1.115 | 0.289 | 0.463 | **0.678** | 4.292 | – |
| ReFL* | 5.903 | 22.975 | 1.195 | **0.298** | 0.470 | 0.656 | 4.299 | 16 |
| DRaFT-LV* | 5.729 | 22.932 | 1.178 | 0.296 | 0.458 | 0.636 | 4.236 | 24 |
| DanceGRPO | 6.022 | 22.803 | 1.218 | 0.297 | 0.414 | 0.585 | 4.353 | 480 |
| Direct-Align | 6.032 | 23.030 | **1.223** | 0.294 | 0.448 | 0.668 | **4.373** | 16 |
| SRPO | **6.194** | **23.040** | 1.118 | 0.289 | **0.505** | 0.665 | 4.275 | **5.3** |

Table 1: **Comparison of Online-RL methods on automatic Evaluatio on the HPDv2 Benchmark. * indicates code implement by us.**

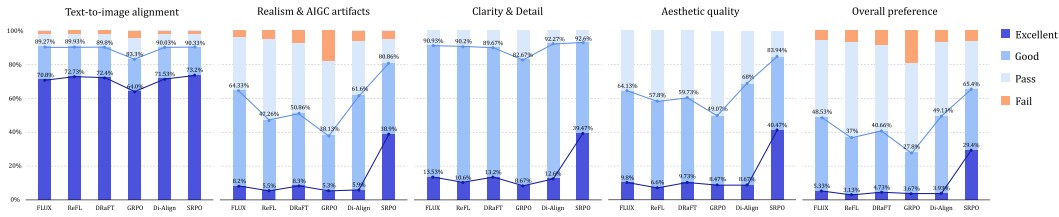

Figure 4: **Comparison of human evaluation results for Vanilla FLUX, ReFL, DRaFT_LV, DanceGRPO, Direct-Align, and SRPO** on HPDv2 benchmark. SRPO demonstrates significant improvements in Aesthetics and achieves a substantial reduction in AIGC artifacts.

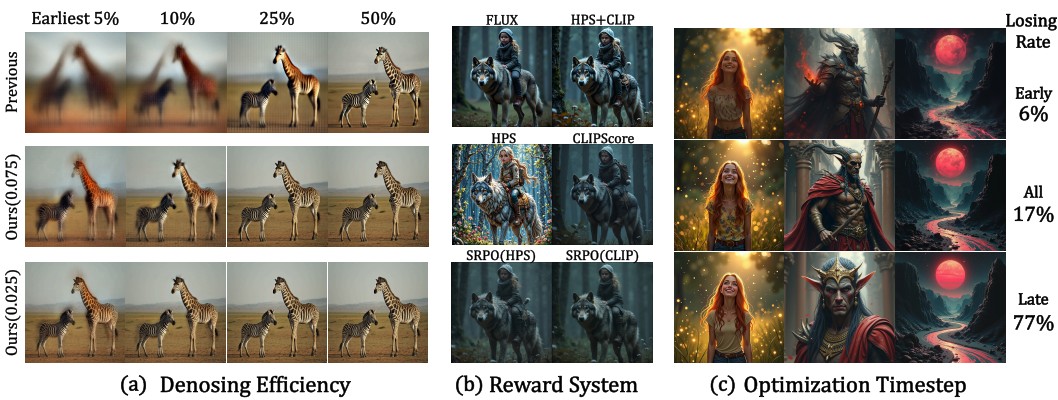

Figure 5: **Ablation study on Denoising Efficiency, Reward System, and Timestep Optimization.**

## 4.2 MAIN RESULT

**Automatic Evaluation Results.** Our method demonstrates three key advantages when train with HPSv2.1 (Table. 1): (1) immunity to HPS score inflation from overfitting, (2) superior performance across multiple reward metrics compared to SOTA methods, and (3) 75× greater training efficiency than DanceGRPO while matching or exceeding all Online-RL baselines in image quality. To further support that our method avoids overfitting to the reward, we present additional results in the Appendix, where models are finetuned with different rewards and consistently show no color cast, oversaturation, or other reward hacking artifacts (see fig. 8).

**Human Evaluation Results.** Our method achieves state-of-the-art (SOTA) performance, as shown in fig. 4. Methods that directly optimize for reward preferences, including Direct-Align, demonstrate suboptimal performance in terms of realism, even falling short of the baseline FLUX model due to reward hacking. In fig. 3, we present a visual comparison between DanceGRPO and our method. The full set of model visualizations is provided in the Appendix (see fig. 3). Although DanceGRPO can improve aesthetic quality and achieve relatively high scores after reinforcement learning, it often introduces undesirable artifacts, e.g., excessive glossiness (row 2, column 1) and pronounced edge highlights (row 2, column 6). To further verify the enhancement in realism, we selected the first 200 prompts from the photo category in the dataset. We augmented these prompts by prepending realism-related words for the vanilla FLUX. fig. 7 (b) shows that the direct generation of our main model significantly outperforms FLUX.1.dev involving lighting and realism-related words. In contrast, our SRPO substantially improves FLUX across realism, aesthetics, and overall user preference. To the best of our knowledge, this is the first approach to comprehensively enhance realism in large-scale diffusion models, increasing the excellent rate from 8.2% to 38.9% without requiring additional training data. In addition, as shown in fig. 7 (a), our enhanced FLUX.1.dev through SRPO surpasses the latest open source FLUX.1.krea on the HPDv2 benchmark.

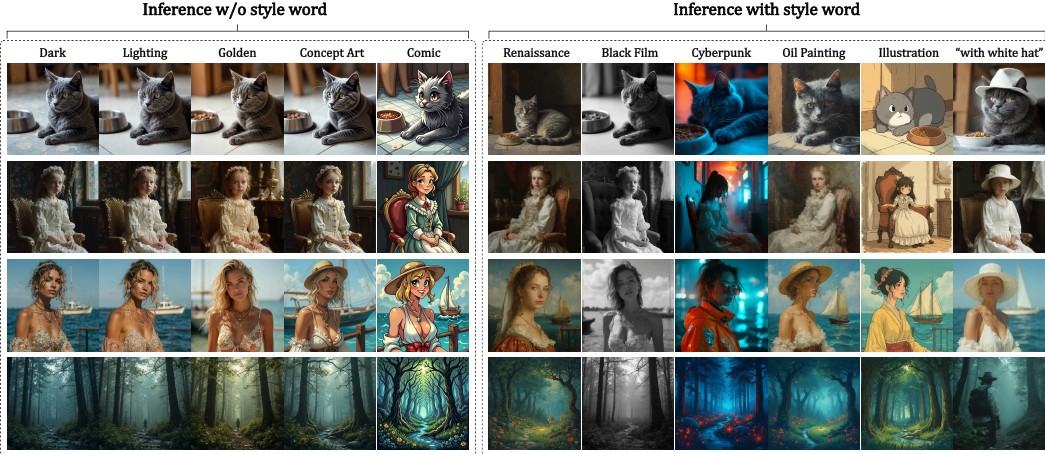

Figure 6: **Visualization of the SRPO-controlled model for different style words.** The *Inference w/o style word* case shows that our method shifts the model's overall output distribution, making most generations stylized even without explicit control words.

## 4.3 ABLATION STUDY

**Denoising Efficiency.** We compare the final images generated by standard one-step sampling (Ho et al., 2020) used in previous method (Xu et al., 2023), which utilize model predictions, with those produced by our method at early timesteps. As illustrated in fig. 5, the standard method still exhibits noticeable artifacts throughout a significant portion of the denoising process. In contrast, Direct-Align, which primarily relies on ground truth noise for prediction, is able to recover the coarse structure of the image even at the initial 5% of timesteps, and produces results that are nearly indistinguishable from the original image at 25%. Furthermore, we investigate the effect of the proportion of model-predicted steps within the total denoising trajectory (as shown in the two rows in fig. 5). The results indicate that a shorter proportion of model prediction leads to clear images. These findings demonstrate the optimization capability of Direct-Align during the early stage.

**Optimization timestep.** We compare three training intervals using Direct-Align without late-timestep discount and PickScore, as shown in fig. 5: Early (the first 25% of noise levels), All (the entire training interval), and Late (the last 25% of noise levels). We randomly selected 200 prompts from the HPD test set for human evaluation. Annotators were asked: *Do any of these three images show hacking artifacts, such as being too saturated, too smooth, or lacking image details? Mark the worst as hacked*. We observe that training exclusively on the late interval leads to a significant increase in the hacking rate, likely due to overfitting to PickScore's preference for smooth images. When training over the entire interval, the hacking rate remains considerable, as this scheme still includes the late-timestep region.

**Effectiveness of Direct-Align.** The core contribution of Direct-Align is its ability to address the limitations of previous methods that only optimize late timesteps. Direct-Align introduces two key components: early timestep optimization and late-timestep discount. In fig. 7 (d), we ablate these components in the Direct-Align. As shown in Eq. 2 and Eq. 5, removing early timestep optimization causes the reward structure to resemble ReFL, leading to reduced realism and increased vulnerability to reward hacking, such as oversaturation and visual artifacts. Similarly, removing the $\lambda(t)$ discount makes the model prone to reward hacking, resulting in oversaturated and unnatural textures. These findings confirm the importance of our approach in overcoming the limitations of late-timestep optimization. fig. 7 (d) also compares the use of inversion versus the direct construction of the reward as in Eq. 8. Although direct construction yields slightly lower texture complexity than inversion, the results remain competitive. These results highlight the potential of the SRPO reward formulation for future applications in other online RL algorithms that are unable to support inversion.

**Fine-Grained Human Preference Optimization.** The key contribution of SRP is its effective guidance of RL direction by manipulating control words. As shown in fig. 6, several control words successfully control the direction of fine-tuning on HPDv2 and HPS, adjusting brightness (columns

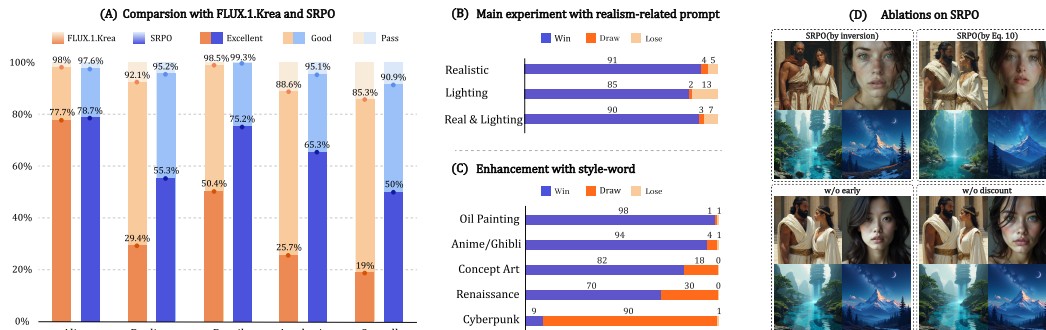

Figure 7: **Overview of experimental results demonstrating the key properties of our SRPO method on the HPDv2 dataset:** A: Comparison between FLUX.1.Krea and FLUX.1.dev tuned by SRPO. B: Comparison between our main model and vanilla FLUX.1.dev using realism-related word. C: Illustration of enhanced style control achieved through the incorporation of style-word conditioning. D: Ablation study on the main components of SRPO.

1–3) or shifting outputs to comic/concept art styles. For challenging styles such as Renaissance, the fine-tuned model cannot always generate the desired style directly; the style word typically needs to be explicitly added during inference to achieve the correct result. We validated this with a user study that compared models before and after training with style words. For user study, we selected the first 200 prompts from the *photo* subclass, as these prompts do not contain explicit style or IP terms. Each prompt was prepended with style words to generate two images for each prompt. Annotators then evaluated each image pair for adherence to the intended style, and in cases of equal style fidelity, overall aesthetics were used as a tiebreaker. As illustrated in fig. 7 (c), our approach enables more effective style control and improves the performance of FLUX on certain styles. However, the degree of improvement depends on the reward model's ability to recognize specific style terms; For the Cyberpunk style, its infrequency in training data makes it difficult for the reward model to recognize this style. Consequently,the overall improvement in human evaluation is limited.

## 5 CONCLUSION

In this work, we propose a novel Online RL framework to align text-to-image (T2I) models with fine-grained human preferences, allowing fine-grained preference adjustment without the need for fine-tuning reward. Our approach addresses two primary limitations of existing methods. First, we overcome the sampling bottleneck, allowing the RL algorithm to be applied beyond the late-stage generation of clean images. Second, we revisit the design of reward signals to enable more flexible and effective preference modulation. Through comprehensive experimental evaluations, we demonstrate that our method outperforms state-of-the-artapproaches in terms of both image realism and alignment with human aesthetic preferences. Compared to DanceGRPO, our framework achieves over a $75\times$ improvement in training efficiency.

**Limitations & Future Work.** This work has two main limitations. First, in terms of controllability, our control mechanism and certain control tokens are somewhat outside the domain of the existing reward model, which may result in reduced effectiveness. Second, in terms of interpretability, since our method relies on similarity in the latent space for reinforcement learning, the effects of some control texts may not align with the intended RL direction after being mapped by the encoder. In future work, our aim is to (1) develop a more systematic control strategy or incorporate learnable tokens and (2) fine-tune a vision language model reward that is explicitly responsive to both control words and the prompt system. Additionally, the SRPO framework can be extended to other online reinforcement learning algorithms. We anticipate that these improvements will further enhance the controllability and generalization capabilities of SRPO in practical applications.

## 6 REPRODUCIBILITY STATEMENT

We have made every effort to ensure that the results presented in this paper are reproducible. To facilitate this, we included the source code (training and inference) with exact hyperparameter configurations of our main model in the supplementary material. Our implementation is built on the opensource framework and opensource dataset (Wu et al., 2023), and models can be trained following the instructions provided within a compatible environment.

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

## A CROSS-REWARD PERFORMANCE

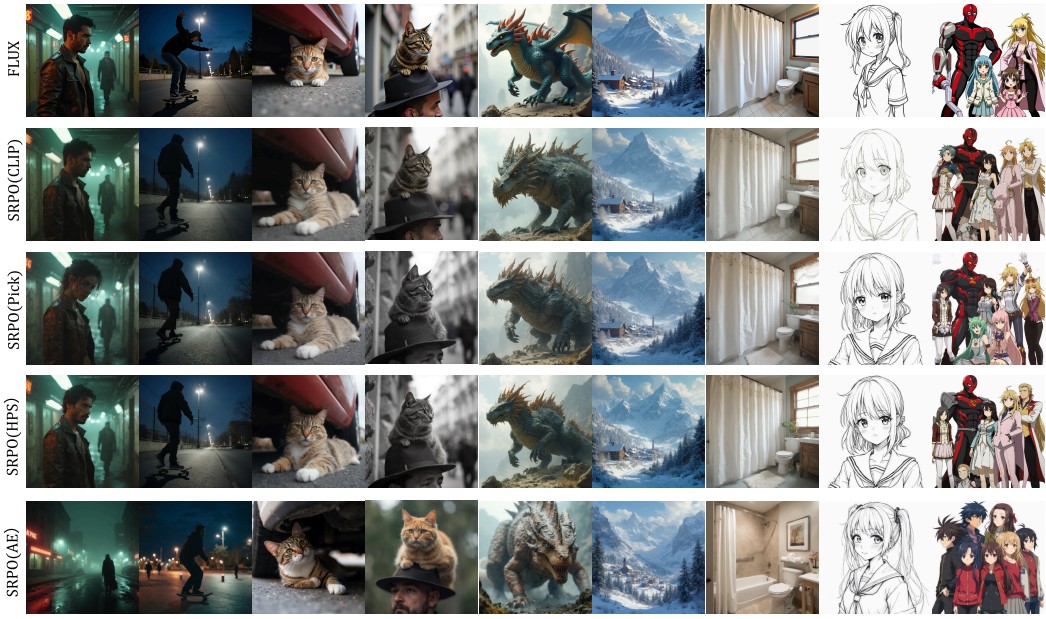

Figure 8: **Cross-reward results of SRPO.**

We evaluated our method using three CLIP-based reward models: CLIP ViT-H/14, PickScore, and HPSv2.1, as illustrated in fig. 8. Our approach consistently enhances image realism and detail complexity across all models, including CLIP, though improvements with CLIP remain limited due to its lack of human preference alignment. Notably, PickScore demonstrates faster and more stable convergence than HPS, while both yield comparable visual quality. Crucially, no reward hacking is observed in our method, highlighting the effectiveness of Direct-Align's design ( fig. 8 (c)) in decoupling optimization from reward-specific biases while preserving alignment with user objectives. Additionally, we validate the generalization of our approach to unimodal rewards. Although SRPO primarily operates on the text branch and thus cannot explicitly control purely image-based aesthetic models, relative reward can still be achieved through data processing techniques. Specifically, we introduce small amounts of noise to the images generated by the model and compute the aesthetic reward for both the noised and the original clean images. This setup naturally forms positive and negative optimization gradients. Although the reward scores for noisy images may not be accurate, they serve to penalize the overall bias in the reward model. Our experiments demonstrate that this approach remains effective in mitigating reward hacking phenomena as shown in fig. 9

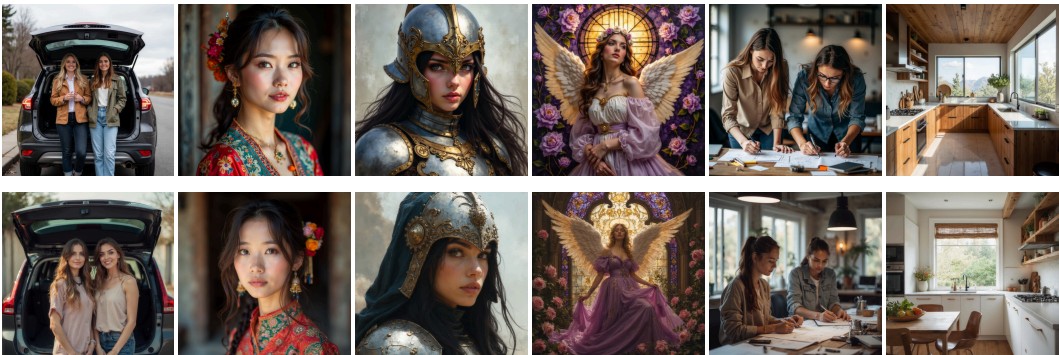

Figure 9: **Extension to the Aesthetic Model.** The first row is trained with Direct-Align using the original Aesthetic Predictor 2.5, while the second row is trained using SRPO.

## B COMPARISON TO GRPO

Our approach is inspired by the group relativity mechanism in GRPO. Similar to GRPO, our method first samples clean images without gradients and then injects noise back into the corresponding intermediate to train. However, our method offers several key advantages over GRPO. First, we apply direct propagation on the reward signal, in contrast to the policy optimization used in GRPO; this leads to significantly improved convergence speed. For example, during FLUX training, we observe that methods based on direct propagation yield noticeable image changes within 30 steps, whereas GRPO requires over 100 steps for comparable results. Second, our approach computes semantic-relative advantages, requiring only a single sample for each update and relying solely on the original ODE. This eliminates the reliance on the diversity of the generative model or sampler. Third, unlike GRPO, which often necessitates additional KL regularization and a reference model to prevent over-optimization, our method directly constrains the optimization by propagating the negative reward signal, thus obviating the need for auxiliary constraints.

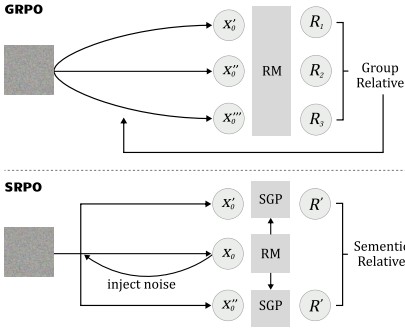

Figure 10: **Comparison on GRPO and SRPO.**

## C HIGH-FREQUENCY WORD STATISTICS IN HPDV2 TRAINING SET

We found that the effectiveness of our method depends on the reward model's ability to perceive control words. Here, we briefly present the word frequency statistics in the HPDv2.1 training set. As discussed in section 4.3, `painting` is the most frequent word and achieves the best experimental results, while the less frequent word `Cyberpunk` yields weaker enhancement effects. Furthermore, we observed that low-frequency words can benefit from being combined with high-frequency words. For example, the *Comic* column in our experiment uses a combination of `anime`, `comic`, and `digital painting`. Similarly, `Renaissance` is constructed by combining `Renaissance-style` and `oil painting`.

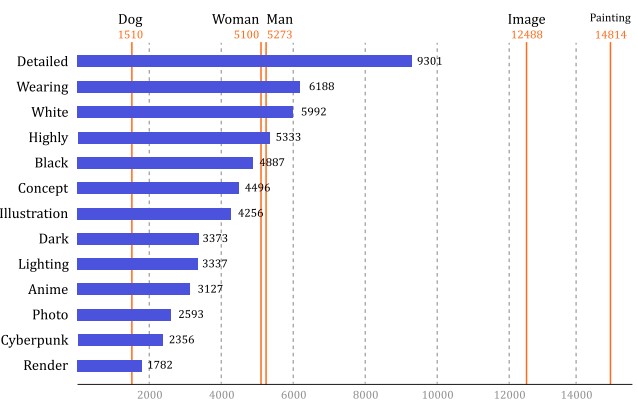

Figure 11: **High-frequency Word Statistics (part) in HPDv2 Training Set.**

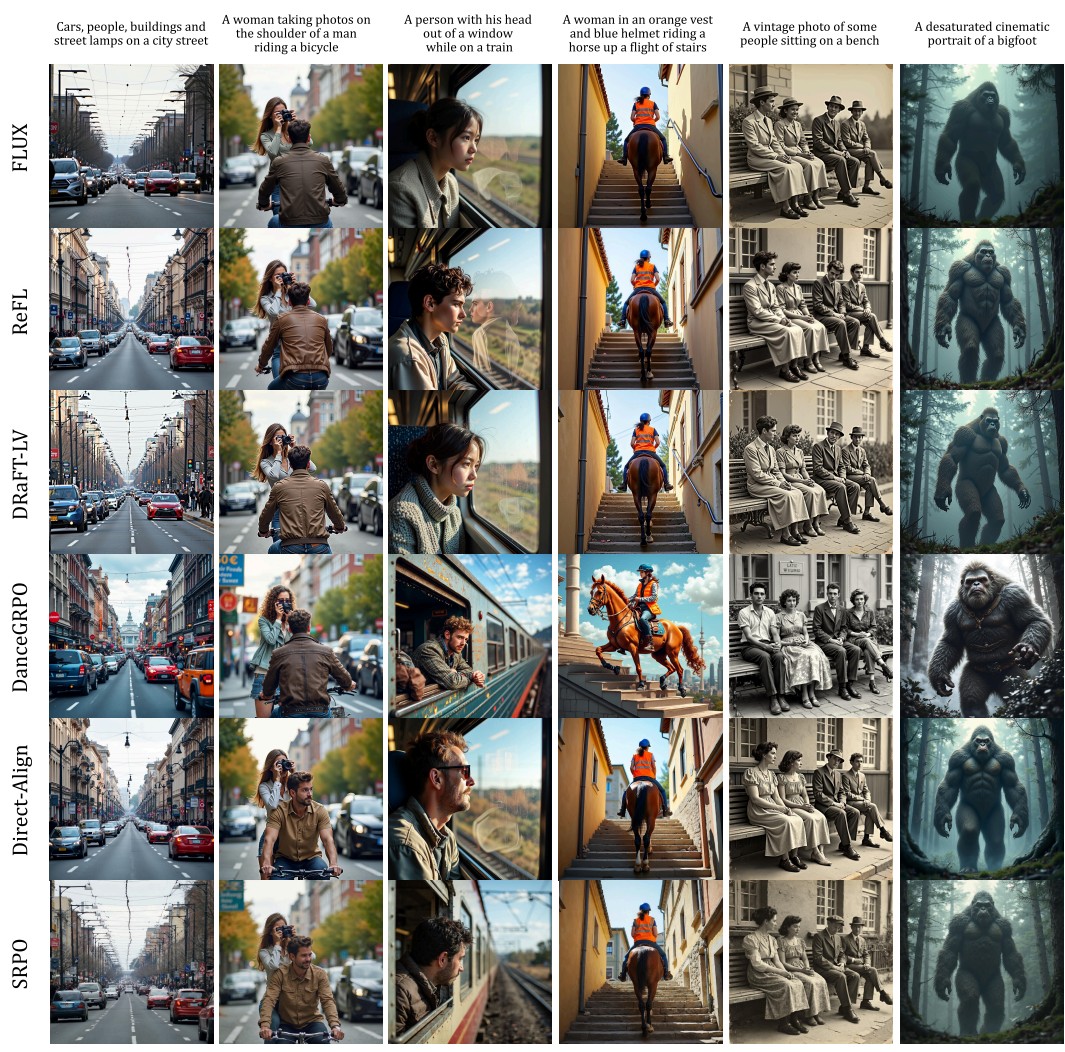

Figure 12: **Qualitative Comparison on several methods with same seed.** Our approach demonstrates superior performance in realism and detail complexity.

## D    ETHICAL STATEMENT

We are committed to the highest standards of ethical research and responsible innovation. In this study, we have carefully evaluated our data, methodologies, and potential applications and, to the best of our knowledge, have identified no significant ethical concerns. All experiments and analyses were conducted in adherence to established ethical guidelines, ensuring the integrity, transparency, and accountability of our research process.

## E    DISCLOSURE ON THE USE OF LLM WITH PAPER WRITING

In the interest of full transparency, the authors declare the use of an LLM-powered language model, GPT-4o, to assist in the writing of this paper. The tool's role was exclusively that of a writing assistant, focused on enhancing the linguistic quality of the text. We wish to emphasize that the LLMs was not involved in any part of the scientific process. All research ideas, methodological choices, data analysis, and conclusions were conceived and executed by the authors. The AI did not generate substantive content, factual information or references. The authors retained full control over the manuscript, critically reviewed all suggestions, and bear complete responsibility for the scientific integrity and accuracy of this work.

Table 2: Evaluation Criteria.

| Criterion | Description | Key Points |
|---|---|---|
| Realism & AI artifacts | Evaluates whether the image looks real and free of AI artifacts | • Whether deformation artifacts appear in the image
• Whether the text is correct (if the image contains text)
• Oily surface or over-saturated colors on objects
• Abnormal highlights on object edges or unnatural transition to background
• Whether the object's texture is overly simple or even |
| Subject Clarity and Detail Complexity | Whether the main subject of the image is clear and detailed | • Whether there is obvious blurriness in the image.
• Whether the main subject of the image is intuitively presented (i.e., not blurry).
• Whether there are any watermarks or garbled text in the image that affect its presentation.
• Whether the texture of the image is complex, for example, whether the texture of leaves is distinguishable.
• Whether the lighting and shadows in the image are prominent, and whether the light source is identifiable. |
| Image-Text Alignment | Measures Image-Text Alignment by grading | • **Excellent:** Over 90% of the elements match the prompt, and the style is fully consistent. If there is text, it should be fully generated and naturally embedded in the image.
• **Good:** 70%–90% of the elements match the prompt. Minor errors in the text are allowed.
• **Pass:** 50%–70% of the elements match the prompt. Most key elements are present, or the image generally looks similar to the prompt at first glance.
• **Failed:** Many key elements are missing, or the style does not match the prompt. |
| Aesthetic Quality | No need to reference the prompt; evaluate the aesthetic appeal of each image based on composition, lighting, color, etc. | • **Excellent:** The image has a strong atmosphere and is highly visually appealing.
• **Good:** The image stands out in at least one aspect—composition, lighting, or color—making it comfortable to view or eye-catching.
• **Pass:** The image has no obvious flaws, but its aesthetic appeal is average.
• **Failed:** The image is unattractive to look at. |
| Overall Quality | Comprehensively evaluate the overall preference for the image. | • **Excellent:** All dimensions are rated as Excellent.
• **Good:** At least half of the dimensions are rated as Excellent.
• **Pass:** No dimension is rated as Failed.
• **Failed:** Any dimension is rated as Failed. |