# OpenReview forum: "Directly Aligning the Full Diffusion Trajectory with Semantic Relative Preference Optimization"
_ICLR.cc/2026/Conference — ICLR 2026 Conference Withdrawn Submission_

### Official Review · Reviewer_ufow · 2025-10-27

**Soundness:** 2
**Presentation:** 1
**Contribution:** 2
**Rating:** 2
**Confidence:** 4

**Summary:**

This paper introduces Direct-Align which is a online reward optimization framework for diffusion models. It identifies that previous online optimization frameworks that directly backpropagates gradients from reward models focus on small number of timesteps closer to the clean data, and proposes to add a predefined noise to a sampled data so that there is a path for perfectly reconstructing the sample. In addition, the paper introduces Semantic Relative Preference Optimization where rewards are formulated with text-conditioned signals which enables dynamic control of the reward model via prompt augmentation. The quantitative results demonstrate this framework achieves the state-of-the-art performanceon HPDv2 benchmark and human evaluation.

**Strengths:**

- The paper tackles a challenging problem of online reward optimization for diffusion models which backpropagtes through the reward models which can potentially train more efficiently than reinforcement learning frameworks that assume black box reward models or Diffusion DPO that requires preference datasets.
- It demonstrates the state-of-the-art performance in a image generation benchmark.

**Weaknesses:**

- Single-step image recovery: it is not clear why ensuring perfect reconstruction makes the learning better. At a high noise level, using the ground-truth noise as a hint would confuse the reward model as it guides the reconstruction towards the original sample. In equation 5, $\Delta \sigma$ is not properly defined, and it's also not clear what $r = r()$ means. Both $\sigma_t$ and $\sigma$ are used and not explained.
 - Reward Aggregation Framework: in line 213, '...decaying discount factor through gradient accumulation, which helps mitigate reward hacking at later timesteps' is not properly explained and it is not clear why this is true. Also $\lambda(t)$ is not properly defined and there is another function $r()$ that seems different from equation 5 and not properly introduced. In line 211, 'sequence of images ${x_k,...,x_{k-n}}$ shares the same subscript as $x_0$ but it appears the subscripts in the sequence denote different samples, which is confusing.
- Inversion-Based Regularization: it is hard for the reviewer to understand what equation 9 entails and a further explanation of this section would be appreciated.
- Section 4.3, Denoising Efficiency: it is expected if using ground truth noise, reconstruction is much easier. It is not clear what this section is adding in value.

Other sources of unclear notations:
- Equation 1 says R and equation 7 says RM which are both not defined.
- Line 232: 'Although ... ' does not complete a sentence
- Equation 9: $r_1$ not defined
- Section 4.1 title: ending with '.'

**Questions:**

It would be great if the points in the weaknesses section can be addressed.

---

### Official Review · Reviewer_cPDX · 2025-10-27

**Soundness:** 2
**Presentation:** 1
**Contribution:** 3
**Rating:** 2
**Confidence:** 3

**Summary:**

The paper introduces Direct-Align and Semantic-Relative Preference Optimization (SRPO) to align text-to-image diffusion models across the full trajectory. Direct-Align pre-samples a noise prior which facilitates one-step denoise/inversion to recover the image from any timestep via interpolation, avoiding costly multistep backprop and enabling stable early-timestep optimization. It targets the common failure where prior methods optimize only late steps and become prone to reward hacking; by unlocking early steps, the paper argues it mitigates this issue. The authors further propose SRPO that makes the reward text-conditioned using positive/negative prompt augmentation, allowing quick online adjustment without repeatedly re-tuning the reward model.  In experiments (e.g., on FLUX.1-dev), the authors report large gains in human-evaluated realism and aesthetics along with strong training efficiency.

**Strengths:**

- Full-trajectory optimization. Direct-Align can recover an image from any timestep using a predefined noise prior. This enables more stable early-step updates while avoiding late-step over-optimization.

- Superior human evaluation. On FLUX.1-dev, the paper reports large gains in human-rated realism and aesthetics versus baseline.

- Flexible reward adjustment. SRPO makes the reward text-conditioned using positive/negative prompt augmentation.

**Weaknesses:**

- Lack of support for attributing reward hacking to late, few-timestep optimization.
Although late-timestep-focused optimization shows reward hacking, and the proposed method focuses on early timesteps and shows reduced hacking in Fig. 5(c), the paper should provide a deeper analysis of why late timesteps lead to undesired reward hacking. Intuitively, late steps emphasize high-frequency detail, which may encourage hacking, but more concrete analysis is needed.

- There is a significant mismatch between the equations and figures (and the released code) for x_0 prediction.

- Why SRPO training is much faster than ReFL and DRaFT-LV? Supposedly SRPO and DirectAlign should be slower than ReFL, as ReFL can skip the full trajectory sampling, while the proposed methods require the full-trajectory sampling to get the noise prior.

- Closeness to ReFL via timestep reweighting.
The method essentially reduces the weight of gradients at early timesteps, so it is roughly a reweighted ReFL that tunes down early-step contributions. How much do early-timestep gradients actually contribute? If late timesteps dominate the gradient direction, the approach collapses to ReFL.

- Reliability of SRP (prompt-based reward).
SRP relies on simple prompt adjustments and differences of positive–negative text embeddings. What direct evidence supports the claim that SRP fixes the oversaturation tendency in HPS-v2.1? Please show targeted ablations and counter-examples.

- Significant ambiguity in implementation details.
First, selecting \Delta\sigma_t is hard; the paper does not clearly state how it is set during training, and it lacks comparisons of different \sigma_t schedules. Second, in the implementation, training uses steps 5–30 of 50, not the full trajectory. Is this correct?


- Writing and notation issues.
The paper is not well written and is hard to understand. The “Inversion-Based Regularization” paragraph is difficult to interpret. Could the authors give more explanation on why inversion is needed and how it's implemented? Notation is unclear and inconsistent. For example, in Eqs. (1–2) r denotes a reward value, but in Eq. (5) it denotes a reward function. In Eq. (6), the mixed use of k and t is confusing.

**Questions:**

1.	In Equation (6), shouldn’t the discounted weight be placed inside the summation?
2.	In Figure 6, is there multiple models tuned, each is tuned with a specific style word applied? And some tuned models can output desired
3.	Why there are degradations in GenEval scores in all RL-aligned models?

---

### Official Review · Reviewer_eat3 · 2025-10-30

**Soundness:** 3
**Presentation:** 2
**Contribution:** 2
**Rating:** 4
**Confidence:** 5

**Summary:**

The paper introduces several contributions aimed at improving reinforcement learning for diffusion models:

1. Mitigating reward hacking by mixing predicted noise with ground-truth noise, enabling more accurate single-step image recovery and more effective gradient propagation with respect to individual denoising steps.
2. Reward aggregation across diffusion steps to reduce reward hacking in later stages of the generation process.
3. Leveraging embedding arithmetic in CLIP-like encoders to counteract negative biases in reward signals.
4. Inversion-based regularization, which builds upon their single-step recovery framework to further stabilize training.

**Strengths:**

1. All proposed components are simple and easy to implement.

2. The method demonstrates improved computational efficiency, it is also evidenced by the GPU-hour comparison in Table 1.

3. The approach effectively leverages existing CLIP-based reward models, showcasing compatibility with widely used and accessible architectures.

**Weaknesses:**

1. While the paper presents a practical and nearly plug-and-play method, it lacks ablation studies to justify key design choices. For example, the effects of the constant K from equation 9, $\lambda(t)$ from equation 6, weighting from equation 5.
2. The writing can be somewhat confusing in places. For instance, in Equation 9,  K is a constant, $\Delta \sigma_t$ should also remain constant, whereas $\epsilon_\theta$ is not. It is unclear how the operation K + $\epsilon_t$ is defined - does it mean adding a constant offset to each predicted noise value?
3. The contributions are not well isolated, making it difficult to discern which specific components of the method are responsible for the observed improvements.
4. The paper relies on a somewhat outdated CLIP-based reward model. It raises concerns about generality — the method should also be validated with more recent state-of-the-art reward models. In particular, CLIP’s limitations (e.g., lack of support for non-square resolutions) may restrict the applicability of the proposed approach.

**Questions:**

1. Could you provide ablation studies for the main design choices in your method? Paper does not include training dynamics and hyperparameters.

2. Could you elaborate further on the proposed inversion-based regularization — particularly its formulation and impact on training stability?

3. Isolating and evaluating each contribution separately would greatly strengthen the paper’s clarity and evidential value.

4. Extending the experiments to include or adapt the method to state-of-the-art reward models would be highly beneficial for future research and community adoption.

---

### Official Review · Reviewer_pJaA · 2025-10-31

**Soundness:** 1
**Presentation:** 2
**Contribution:** 2
**Rating:** 2
**Confidence:** 3

**Summary:**

The paper proposes two components:

**Direct-Align:** skips the slow denoising by using interpolation between noise and the target image, which is faster and avoids over-optimisation.

**SRPO:** updates rewards on the fly using text prompts, eliminating the need for endless offline tuning.

**Strengths:**

## Presentation: ~25th percentile

Structure: The paper follows the most common structural pattern, which aligns with reader expectations.

## Soundness: ~15th percentile

The soundness is supported by the reliability of CLIP.

## Contribution: 20th~50th percentile

The methodology, using a CLIP inner product as a reward, was new to me, but not particularly surprising.

## Note
I hope the AC is aware that the rating is calibrated using percentiles to reduce evaluation noise effectively.

**Weaknesses:**

I don’t think this paper is ready for publication.

## Presentation
There are significant issues in your writing.

* In your abstract, line 27, “by over 3x”, what is the reference point like, in what unit? I list this issue as critical because it is presented in your abstract.
* In Section 3.1, you introduce several mathematical notations without explanation, while I have difficulty corresponding them to your text, such as $\alpha_t, \sigma_t, \Delta \sigma_t, \lambda(t) $. I’m thus having difficulty understanding one of your important equations, Eq. (5). Although the parameter $\alpha_t$ is trivial for readers who are familiar with diffusion models, you ought to either write about them whenever you introduce them (I do not even recommend putting it in the appendix as it breaks the reader’s flow) or explain them with an intuitive alternative presentation.
* My flow of reading the paper was occasionally interrupted by unimportant details presented in the text, particularly when I was reading your introduction and experiment. You could have simplified these details and made room for your mathematical explanation. For example, suggest the most significant work related to each idea in your introduction and leave the others in the related work. The introduction section should be literally the introduction.

## Soundness
1. It’s known that CLIP-based models are CLIP-blinded, e.g., [1], meaning they are imperfect for less text-expressible tasks. The paper could have explored DINOv2, an embedding model purely based on images, but the comparison, which could have strengthened this paper's contribution, is also missing.
2. I remain doubtful about the effectiveness of this method. It does not completely eliminate the null hypothesis:
  * In Table 1, the standard deviation is missing; to my empirical experience, the standard deviation is roughly Aes: $10^{-0.4},$ Pick: $10^1,$ IR: $10^{0.8},$ HPS: $10^{-1.5}$. I recall that algorithms like DiffusionDPO and DDPO can easily surpass one standard deviation, but such a baseline is not included.
  * Most of the quantitative results in your main paper, other than Table 1, are based on subjective tests. I keep doubting these tests because 1. they are not reproducible on the same group of people, 2. it’s not very clear about the distribution of image choice, 3. human evaluations are loudly noisy. I would rather believe in the number written as a statistical fact, extensively evaluated on visual langauge models, which are biased but sufficiently reliable compared to humans.

## Contribution:

The idea is new to me, but not very surprising, given that replacing GRPO with $f_{\mathrm{img}} (\mathbf{x})^\top \mathrm{C} $. A potential advantage over GRPO is that it enforces linearity, but the potential exploitation I had expected was missing in the paper.

The image embedding can be free from text-based; you could consider introducing DINOv2 for deeper contribution.

[1] Adejumo, Abduljaleel, et al. "A Vision Centric Remote Sensing Benchmark.




Because I was unable to understand one of your core components, Eq. (5), I rate your contribution at the lower bound 20th percentile of my estimation range, but I fairly lower my confidence to 3.

The final rating was simply the mean of the estimated percentile.

**Questions:**

See the weaknesses.

---

### Note · Authors · 2025-11-13

I have read and agree with the venue's withdrawal policy on behalf of myself and my co-authors.